# Chinese Resident Preferences for African Elephant Conservation: Choice Experiment

**Shuokai Wang [1], Zhen Cai [2]** , **Yuxuan Hu [3], Giuseppe T. Cirella [4]** and **Yi Xie [3],***

1   School of Economics, Henan University, Kaifeng 475001, China; wangshuokai94@gmail.com
2   Center for Agroforestry, University of Missouri, Columbia, MO 65211, USA; caiz@missouri.edu
3   School of Economics and Management, Beijing Forestry University, Beijing 100083, China; yuxuan_hu@bjfu.edu.cn
4   Faculty of Economics, University of Gdansk, 81-824 Sopot, Poland; gt.cirella@ug.edu.pl
*   Correspondence: yixie@bjfu.edu.cn; Tel.: +86-10-6233-7020; Fax: +86-10-6233-7674

**Abstract:** Despite passionate efforts to preserve African elephants worldwide, their numbers continue to decline. Some conservation programs have suspended operations because the funds provided by various governmental and non-governmental organizations (NGOs) cannot cover the enormous expenses of countering poaching, habitat destruction, and illegal ivory trading. This study investigates Chinese resident preferences for African elephant conservation using a choice experiment model. Results indicated that two-thirds of our 442 respondents with relatively higher education and income levels were willing to donate to conserve African elephants. Respondents were willing to donate RMB 1593.80 (USD 231.65) annually to African elephant conservation. Chinese residents were willing to donate the most to anti-poaching RMB 641.25 (USD 93.20), followed by enhancing habitat quality RMB 359.07 (USD 52.22), combating the illegal trade in ivory RMB 355.63 (USD 51.69), and alleviating human–elephant conflicts RMB 237.85 (USD 34.57). Our results suggest that accepting public donations could be an efficient way for NGOs to better preserve African elephants.

**Keywords:** African elephant; transnational conservation; donation; public opinion; China

## 1. Introduction

African elephant conservation is one of the most challenging issues in global biodiversity efforts [1–9]. The number of elephants in Africa has plummeted over the past 25 years; approximately 111,000 have been killed between 2006 and 2015 [10]. This alarming trend indicates that current strategies in preserving African elephants, targeting the threats of habitat loss, human-wildlife conflicts, poaching, and illegal trading of ivory [11–13] may need to be reinforced. Current efforts to protect African elephants are conducted by 37 African nations. Those states protect the elephants and their habitats using different management strategies, including: designating areas occupied by elephants as protected [14,15], implementing integrated conservation and development projects to alleviate human–elephant conflicts [16,17], enforcing laws against poaching, and educating the public about the disastrous consequences of the illegal ivory trade [18]. From 2010 to 2017, African nations received an estimated USD 500 million in donations from governmental and non-governmental organizations (NGOs) mainly from developed countries to improve law enforcement in protected areas [19] and approximately USD 8 million annually in donations to improve public awareness of illegal trading of African wildlife species [20,21]. As one of the largest developing countries, China is embarking on contributing to African wildlife conservation through government initiatives. Developing countries have also started to support African elephant conservation by providing financial assistance in recent years. For example, in 2015, China donated USD 1.7 million worth of goods

for wildlife conservation in Botswana as part of its pledge of USD 10 million to support wildlife conservation throughout Africa [22]. However, financial assistance from governments and NGOs is limited and is not enough to conserve Africa's elephant populations efficiently. Gray and Gauntlett (2017) estimated that an additional USD 600 million annually, more than ten times the current level of investment in African elephant conservation, is needed to enforce Africa's anti-poaching laws [19]. Implementation of the National Elephant Action Plan through 2030, adopted by 19 African nations as the principal guide for elephant conservation, will require more than USD 1 billion [23]—most of the participating governments are not able to fund this budgetary requisite. As a result, the lack of funds has restricted some states from participating in elephant conservation. For instance, in 2019, Botswana announced that it would no longer ban elephant trophy hunting due to lack of funds in addressing human–elephant conflicts [24]. Therefore, extending the funds available to African nations to preserve elephants is one of the most urgent missions faced by the global conservation community.

African elephant conservation needs public support from both African countries and others from across the world. Conservation funds tend to flow from wealthier countries to poorer ones [20]. Moreover, public support of elephant conservation and individual donations are much more common in developed countries than in developing countries [25]. Studies have shown that public efforts in domestic wildlife conservation are increasing significantly in developing countries [26]; however, knowledge on public awareness and willingness-to-donate (WTD) to wildlife conservation are limited. African elephants are one of the most well-known transnational wild animals in China, with a long history of ivory carving culture and industry [27]. Chinese residents are familiar with ivory products (e.g., carved high-value works of art, jewelry, and piano keys) and their market values. Chinese residents' awareness of the non-use value of African elephants has been improved as a result of an increase in wildlife education-based campaigns [28,29]. For example, recent media campaigns in China have effectively improved public awareness of the brutality of elephant poaching by using graphic images of bloody tusks and bodies of elephants rotting in the heat [26]. China has executed a ban on commercially utilizing and processing elephant ivory and derivative products since 2016 [27]; however, the public's WTD to African elephant conservation is unknown and relevant literature is limited. The only study, according to our knowledge, was by Wang et al. (2018), who investigated the potential for Chinese public donations to African elephant conservation using a contingent valuation model [29].

This study aimed to explore the potential of public donations to African elephant conservation in developing countries by surveying Chinese residents' WTD for African elephant conservation. In specific, this study used a choice experiment method to estimate Chinese residents' preferences of four current African conservation strategies, including: anti-poaching efforts, habitat enhancement, alleviation of human–elephant conflicts, and dismantling of the illegal trade in ivory. This method can also help us identify and rank Chinese residents' WTD to each of the four conservation strategies selected. To the best of our knowledge, this is the first time that the choice experiment method has been applied in this context in China.

## 2. Method

### 2.1. Choice Experiment Design

The choice experiment method is based on random utility theory, which assumes that rational individuals, when presented with several choices, select the option that maximizes utility. Individual utility is determined by attributes and characteristics of the goods presented [30], which allows for the estimation of the economic value of each specific attribute using individuals' willingness-to-pay (WTP) for that attribute (i.e., a willingness to accept it). WTD is one specific form of WTP. The choice experiment method is frequently used to estimate the value of biodiversity [31], especially since its ability to measure non-use values must be taken into account [32]. The choice experiment has been tested to estimate WTP for transnational biodiversity protection in cross-country settings in Denmark and Sweden [33], habitat conservation in the Baltic Sea region [34], and migratory bird conservation

in Denmark and The Netherlands [35]. Moreover, choice experiments have also been used when estimating the economic value of environmental services such as carbon sequestration that have passive or non-use values [36–38].

In this study, the choice experiment was designed to look at Chinese residents' preferences in terms of donating to elephant conservation in Africa. Public decision-making issues are typically multi-dimensional in which dimensions are considered as inter-dependent [39]. To identify potential attributes that may affect Chinese residents' WTD to elephant conservation, we interviewed researchers and administrators from wildlife conservation programs in Africa and an extensive literature review on the topic was performed. Utilizing Hoyos' (2010) research [40], we correspondingly linked attributes and donation levels to best satisfy and correspond with the problem, and reviewed the choice experiment's design (i.e., pilot test) with a number of conservationists and experts in the field before administering the choice experiment.

Table 1 presents the four conservation strategies selected as attributes for the study: (1) improvement of habitat quality, (2) anti-poaching, (3) alleviation of human–elephant conflicts, and (4) combating illegal ivory trade. The fifth attribute was the annual WTD to African elephant conservation made up of three assigned levels: RMB 50 (USD 7.27), RMB 100 (USD 14.53), and RMB 500 (USD 72.67) (Exchange rate used is USD 1 = RMB 6.88, sourced from Bloomberg Currency Exchange at: https://www.bloomberg.com/quote/USDCNY:CUR (accessed 12 December 2018)). The levels of donation presented to participants were based on prior studies of public WTP for wildlife conservation in China and discussions with environmental economic experts (e.g., [41]).

**Table 1.** Attributes and levels used in the choice sets.

| Attribute | Level | Description of Level |
|---|---|---|
| Donation | 0 (current)<br>RMB 50 (USD 7.27)<br>RMB 100 (USD14.53)<br>RMB 500 (USD 72.67) | Annual donation for the support of African elephant conservation |
| Habitat | Yes<br>No (current) | Improvement of habitat quality |
| Poaching | Yes<br>No (current) | Anti-poaching |
| Conflict | Yes<br>No (current) | Alleviation of human–elephant conflict |
| Illegal trade | Yes<br>No (current) | Combating illegal ivory trade |

A fractional-factorial design was used to generate choice questions. SAS v9.4 software and an experimental design [42,43] were utilized for the study. In total, 32 African elephant conservation management practices with a D-efficiency were obtained. Of the three management practices, one choice set for each was formulated—including a status quo option. Respondents were asked to choose one option from the choice set (Table 2). Each questionnaire contained eight choice sets to ensure effectiveness of the research [44]. To avoid potential respondent fatigue in answering the choice experiment questions, two survey versions were administrated, each designed with 16 African elephant conservation management practices. The status quo alternative was used to estimate their WTP donation value in accordance with standard demand theory [42,45].

**Table 2.** Sample choice experiment presented in the questionnaire.

| Question | | Management Practice A | Management Practice B | Management Practice C |
|---|---|---|---|---|
| Donation for support of African elephant conservation annually | | RMB 50 | RMB 100 | 0 |
| *Management strategy used* | Improvement of habitat quality | Yes | No | No |
| | Anti-poaching | Yes | No | No |
| | Alleviation of human–elephant conflicts | No | Yes | No |
| | Combating the illegal ivory trade | No | Yes | No |

The choice experiment questions started with a hypothetical scenario, in which it is assumed that there is an NGO that cares for an African elephant conservation project. The project has many management practices, with the only differences among practices being the management strategies and donation value. The four management strategies used in the choice experiment model were explained to the respondents. The organization states its collected donations (i.e., from the public) would go directly to their project and would fund its conservation activities. Hypothetical bias was also mentioned in the questionnaire. As such, voluntary donations as the payment vehicle in contingent valuation (CV) may lead to a hypothetical bias by which the tendency to overstate one's WTP due to the so-called "warm glow" effect is associated with giving to a good cause [46]. The presence of hypothetical bias has been confirmed by empirical results in several cases [47,48]. However, a meta-analysis found that the CV estimates from a study of welfare measures were somewhat lower than values generated using a revealed-preference method, indicating the external validity of CV estimates [49]. In terms of choice experiments, Carlsson and Martinsson (2001) maintained that they are a viable method for assessing preferences because they found that respondents did not tend to overstate their WTP in a hypothetical setting [46]. To eliminate possible hypothetical bias, respondents were asked to answer the choice experiment questions as if they were actually going to donate.

### 2.2. Data Collection

The questionnaire was made up of five parts. The first part asked for respondent knowledge about and perceptions of biodiversity conservation. The second part gathered information about their understanding of the current status of African elephants and the major threats to their survival. The third part described the current situation of African elephant population, and the fourth, the primary conservation strategies used to protect them. Choice experiment questions were included in part four. The final part of the questionnaire collected information on socioeconomic and demographic information. Four cities were selected as part of the research area, including: Beijing, Guangzhou, Fuzhou, and Shanghai. These cities have many certified (i.e., legal) ivory carving factories, wholesale sites, and retail stores [9]. Responses were collected through Wenjuanxing, an online survey collection agency in China. This agency has a pool of online panelists in China recruited using social media outlets. Wenjuanxing has been used extensively in China to collect data regarding resident know-how of and attitudes about environmental protection and willingness to contribute to it (e.g., [50]). Online studies tend to have samples with higher educational and income levels than the general population [51]. Moreover, online surveys are cost effective in China. The questionnaire was pretested in December 2016 and formally administered in March 2017.

### 2.3. Model Specifications

The choice experiment model is conceptualized using random utility theory, in which a respondent makes a discrete choice by maximizing the utility gained from each option based on the option of each individual attribute [30].The mixed logit model (MLM) was used to analyze our data [52]. The MLM allows for inter-dependence between observed and unobserved components of participant utility and

captures preference heterogeneity by analyzing variations in their WTP for various attributes [40]. The utility function of the MLM uses Equation (1) for the analysis.

$$U_{njt} = \beta' x_{njt} + \delta_{njt} + \varepsilon_{njt} \tag{1}$$

where: $\delta_{njt}$ is a random term with a mean of zero with a distribution over the individual and alternative options pending the underlying parameters and observed data relating to the $n^{th}$ respondent, the $j^{th}$ alternative, and the $t^{th}$ choice. The coefficients represented by $\beta'$ in the MLM are specified as randomly distributed and specific to each other. These models allow one to determine which variables require distributions and the best parameters for the distributions. In most cases, all non-price variables are randomized and price is retained as a non-random variable [53]. Some applications only randomize the cost/price variable [54]. We follow the former convention, since under that approach the distribution of marginal WTP for an attribute is simply the attribute's parameter estimates, which allows the price/cost variable to be restricted to non-positive values for all individuals [55].

In addition to the attribute-only model (model 1), we generated another MLM (model 2), which incorporates interaction variables for selected respondent socioeconomic and demographic characteristics [56,57]: age, gender, income level, and attitude regarding biodiversity conservation [29]. Including interactions with such characteristics can improve the model's accuracy by accounting for its otherwise unobserved effects [58]. The interaction variables were created by multiplying the values of the original choice attribute variables by values for the characteristics [59]. A respondent's WTD for attribute $z$ in model 1 can be calculated using Equation (2).

$$WTD_z = -\frac{\beta_z}{\beta_{donation}} \tag{2}$$

where: $\beta_z$ and $\beta_{donation}$ represent the coefficients of the $z^{th}$ attribute and the donation attribute, respectively.

We also calculated uptake rates, which indicate how the probability of choosing an option changes as the level of that option's attribute changes [60]. We analyzed changes in the probability of a respondent choosing the no-conservation option, i.e., the status quo, in response to a one-unit change to one of the funds or strategy attributes using Equation (3).

$$p_{i,j} = p_i - p_j = \frac{e^{\beta' x_{nit}}}{\sum e^{\beta' x_{nht}}} - \frac{e^{\beta' x_{njt}}}{\sum e^{\beta' x_{nht}}} \tag{3}$$

where: $p_{i,j}$ is the logit probability of choosing alternative $i$ over alternative $j$, $p_i$ is the probability of choosing alternative $i$, $p_j$ is the probability of choosing alternative $j$, $x_{nit}$ and $x_{njt}$ respectively denote the attribute variables of the $i^{th}$ and $j^{th}$ alternatives, and $x_{nht}$ denotes the attribute variable of each alternative.

## 3. Results

### 3.1. Demographics

In total, 442 responses were collected. Table 3 compares the demographic information with the national data, as data from the four cities selected are not available. Our sample was characterized by more women, younger people, a higher education level, and wealthier people than in China's general population. Approximately, 25.12% of the respondents were from Beijing, 25.12% from Fuzhou, 24.88% from Shanghai, and 24.88% from Guangzhou.

**Table 3.** Summary of participant demographic information and China's population from 2017 census data.

| Characteristic | Survey Sample | China |
|---|---|---|
| Gender | | |
| Male | 46.45% | 51.17% [a] |
| Female | 53.55% | 48.83% [a] |
| Age | | |
| Younger than 20 | 5.69% | 24.10% [a] |
| Between 21 and 30 years old | 44.31% | 17.14% [a] |
| Between 31 and 40 years old | 36.26% | 16.14% [a] |
| Between 41 and 50 years old | 9.95% | 17.28% [a] |
| Between 51 and 60 years old | 1.90% | 12.01% [a] |
| Older than 60 | 1.90% | 13.32% [a] |
| Education | | |
| No School | 0 | 5.00% [a] |
| Primary only | 0.47% | 28.75% [a] |
| Middle school | 3.79% | 41.70% [a] |
| High school | 23.93% | 15.02% [a] |
| Bachelor's degree | 60.19% | 14.10% [a] |
| Master's degree | 9.48% | 0.55% [a] |
| Doctoral degree | 1.18% | 0.10% [a] |
| Annual household income | | |
| Less than RMB 30,000 (USD 4360) | 6.16% | 0.02% [b] |
| Between RMB 30,001 (USD 4360) and RMB 80,000 (USD 11,628) | 16.11% | 36.98% [b] |
| Between RMB 80,001 (USD 11,628) and RMB 150,000 (USD 21,802) | 27.96% | 34.00% [b] |
| Between RMB 150,001 (USD 21,802) and RMB 300,000 (USD 43,604) | 37.68% | 15.64% [b] |
| Between RMB 300,001 (USD 43,604) and RMB 1,000,000 (USD 145,348) | 10.66% | 10.71% [b] |
| Greater than RMB 1,000,000 (USD 145,348) | 1.42% | 2.65% [b] |
| Importance of environmental protection | | |
| Extremely Important | 25.12% | N/A |

[a] National Bureau of Statistics of the People's Republic of China, Chinese Statistical Yearbook. http://www.stats.gov.cn/tjsj/ndsj (accessed 12 December 2018). [b] Daily Financial Network. https://mip.mrcjcn.com/n/254734.html (accessed 12 December 2018).

### 3.2. Respondent Attitudes Regarding Donations to Elephant Conservation

Approximately 66.67% of our respondents indicated that they were willing to donate to African elephant conservation. In terms of donation value, 21.69% of this study's respondents selected RMB 50/year (USD 7.27/year), 22.14% selected RMB 100 (USD 14.53/year), and 22.05% selected RMB 500 (USD 72.67/year). Approximately 33.33% of our respondents believed alleviating human–elephant conflicts is the most important. Slightly fewer respondents (i.e., 32.94%) prioritized anti-poaching measures over thwarting illegal trading in ivory and improving the quality of elephant habitats. One-quarter of the respondents (i.e., 25.12%) reported they strongly supported biodiversity conservation and were willing to provide financial support.

### 3.3. WTD to Conservation Strategies

Attribute coefficients, WTD values, and uptake rates were estimated using STATA 15. Table 4 presents the coefficients, standard errors, and odds ratios from regressions of the mixed logit models.

**Table 4.** Results from the mixed logit regression models.

| Attributes | Model 1 (*n* = 10,128) | | | Model 2 (*n* = 10,128) | | |
|---|---|---|---|---|---|---|
| | Coef. | Se. | Odds Ratio | Coef. | Se. | Odds Ratio |
| Donation | −0.0011 *** | 0.0001 | 0.9989 | −0.0025 *** | 0.0005 | 0.9975 |
| Habitat | 0.4120 *** | 0.0488 | 1.5099 | 0.4123 *** | 0.0487 | 1.5103 |
| Poaching | 0.7358 *** | 0.0498 | 2.0872 | 0.7397 *** | 0.0500 | 2.0952 |
| Conflict | 0.2729 *** | 0.0515 | 1.3138 | 0.2765 *** | 0.0517 | 1.3186 |
| Illegal trade | 0.4081 *** | 0.0503 | 1.5039 | 0.4097 *** | 0.0503 | 1.5063 |
| Variables interacted with social and economic factors | | | | | | |
| Donation_age | | | | −0.0007 | 0.0007 | 0.9993 |
| Donation_gender | | | | −0.0003 | 0.0002 | 0.9997 |
| Donation_income | | | | 0.0003 *** | 0.0001 | 1.0003 |
| Donation_attitude | | | | 0.0007 *** | 0.0003 | 1.0007 |
| Donation_Beijing | | | | 0.0008 *** | 0.0003 | 1.0008 |
| Donation_Fuzhou | | | | 0.0007 *** | 0.0004 | 1.0007 |
| Donation_Shangai | | | | 0.0002 | 0.0003 | 1.0002 |
| Derived standard derivation of parameter distribution | | | | | | |
| Habitat | 0.4981 *** | 0.0730 | N/A | 0.4928 *** | 0.0737 | N/A |
| Poaching | 0.5076 *** | 0.0690 | N/A | 0.5131 *** | 0.0691 | N/A |
| Conflict | 0.5801 *** | 0.0700 | N/A | 0.5826 *** | 0.0701 | N/A |
| Illegal trade | 0.5542 *** | 0.0699 | N/A | 0.5543 *** | 0.0702 | N/A |
| Model Characteristics | | | | | | |
| Log-likelihood | | −3230.11 | | | −3220.62 | |
| Pseudo-R$^2$ | | 0.117 | | | 0.128 | |

*** denotes statistical significance at the 1% level.

The mixed logit models were both statistically significant, as *p* values from likelihood ratio tests were both less than 0.001. In model 1, the variables for the conservation strategies have positive coefficients, suggesting the inclusion of each of the strategy's attributes positively affects respondent choice, ceteris paribus. The inclusion of anti-poaching strategy in African elephant management practice increased respondents' preferences for African elephant conservation by 208.7% compared to the practice without anti-poaching. The coefficient of the donation variable is negative, suggesting higher donation values decrease respondents' preferences for African elephant conservation. In model 2, the conservation strategy variables also have positive coefficients. It was found that age, gender, and residing in Shanghai did not have significant effects on respondents' preferences for African elephant conservation. Respondents with higher income levels were more likely to donate to an African elephant conservation project compared to respondents with lower income levels. Support for biodiversity conservation positively affects respondent preference for African elephant conservation. Respondents from Beijing and Fuzhou were more likely to donate to the African elephant conservation program than respondents from Guangzhou.

*3.4. Willingness-to-Donate for African Elephant Conservation*

Respondents' WTD for elephant conservation for each conservation strategy attribute was also calculated. The results of model 1 indicate that, on average, respondents were willing to donate RMB 641.25 (USD 92.90) annually for anti-poaching. Model 1 also shows that respondents were willing to donate RMB 359.07 (USD 52.02) annually, on average, to improve the quality of elephant habitat, RMB 355.63 (USD 51.52) annually to combat illegal ivory trading, and RMB 237.85 (USD 34.46) annually to alleviate human–elephant conflicts (Table 5).

**Table 5.** Willingness to pay for conservation attributes.

| Attributes | Model 1 [†] |
|---|---|
| Habitat | RMB 359.07 (251.65,466.49) |
| Poaching | RMB 641.25 (496.72,785.78) |
| Conflict | RMB 237.85 (141.53,334.18) |
| Illegal trade | RMB 355.63 (247.30,463,95) |

[†] 95% confidence internals are presented in brackets.

### 3.5. Simulation Preferences under Scenarios

Table 6 presents the estimated uptake rates, which measure the changes in participants' willingness to support elephant conservation (i.e., in the form of donations) in response to changes in the funding and conservation strategy attributes. It was found that increasing the donation level from the status quo of zero to the high level of RMB 500 decreases respondents' willingness to donate by 27.93% in model 1 and 57.41% in model 2. A smaller increase, from zero to the medium level of RMB 100, decreases the willingness to donate by 5.73% in model 1 and 12.99% in model 2. Not surprisingly, the smallest increase, from zero to the low level of RMB 50, decreases willingness to donate the least, by 2.87% and 6.52% in model 1 and model 2, respectively.

**Table 6.** Simulated uptake rates for changes in the funding and strategy attribute levels.

| Attribute Change | Model 1 [†] | Model 2 [†] |
|---|---|---|
| | Donation increase | |
| RMB 0 to 500 | −27.93% (−33.57%, −22.28%) | −57.41% (−73.09%, −41.73%) |
| RMB 0 to 100 | −5.73% (−6.95%, −4.51%) | −12.99% (−17.59%, −8.40%) |
| RMB 0 to 50 | −2.87% (−3.48%, −2.25%) | −6.52% (−8.85%, −4.19%) |
| | Conservation strategy | |
| Habitat Improvement | 20.31% (15.73%, 24.89%) | 20.33% (15.75%, 24.91%) |
| Combating poaching | 35.22% (30.94%, 39.49%) | 35.39% (31.11%, 39.68%) |
| Alleviating conflicts | 13.56% (8.61%, 18.51%) | 13.74% (8.78%, 18.71%) |
| Combating illegal trade | 20.13% (15.39%, 24.85%) | 20.20% (15.47%, 24.94%) |

[†] 95% confidence internals are presented in brackets.

We found little difference in the uptake rates between model 1 and model 2 for the prioritization of a conservation strategy. Willingness to donate is highest for anti-poaching measures, with an increase of about 35% from the status quo. Habitat improvement increases support by about 20%, alleviating conflicts by about 13%, and combating illegal ivory trading by about 20%.

## 4. Discussion

This study reveals that many Chinese residents are willing to contribute to African elephant conservation. Since the participants in the study have somewhat greater levels of income and education than China's population overall, we cannot conclude that most people in China are willing to donate to support elephant conservation but can conclude that relatively well-educated and middle to upper income sectors of the population are. Our estimates of average WTD to support elephant conservation are higher than ones reported by Wang et al. (2018) [29], *viz.*, 66.67% vs. 53.36%, likely due to the differences in the levels of education and income between the two samples.

Our results reveal that Chinese residents strongly support wildlife conservation in other countries. Education and outreach efforts in China in recent years helped increase residents' awareness of the plight of African elephants, and further influence their attitudes about illegal ivory products, leading to a reduction in demand of ivory products [61,62]. It further suggests that the average amount of an annual donation to elephant conservation rises with income. Participants in the study preferred to donate amounts that would not have significant effects on their financial situations, but the donations

to conservation would still be considerable given the number of people who fall into higher education and income levels. According to China's census data, roughly 15% of the country's population has at least a bachelor's degree, and according to our results, approximately two-thirds of those individuals would be willing to donate, generating a pool of roughly 100 million potential donors. Assuming only 10% of those people (i.e., 10 million) actually make a donation, the total contributed annually to conserve African elephants would be about RMB 10.58 billion (USD 1.51 billion). Since the donations represent a relatively small portion of one's family income, there is potential for even larger donations, particularly from wealthier households. The mean total annual donation for this group shows a significant contributive amount, which is comparable with the results of Wang et al. (2018) [29], estimating an aggregate WTD ranging from RMB 16.31 billion (USD 2.37 billion) to RMB 42.83 billion (USD 6.23 billion).

Our findings point to several approaches for convincing more Chinese residents to contribute to African elephant conservation. Of the four strategies used in this study, the anti-poaching strategy received the greatest WTD, followed by improving elephant habitat, combating illegal ivory trading, and alleviating human–elephant conflict. Recent media campaigns in China have been effective in making the public aware of the brutality of elephant poaching using graphic images of bloodied tusks and bodies of elephants rotting in the heat. Participants' willingness to donate to anti-poaching measures suggests that much of the motivation is related to the concern for animal welfare. Hence, programs designed to thwart poaching are likely to appeal strongly to individuals in China who have relatively higher education and income levels. Programs can aim to combat illegal ivory products by linking them to poaching. Other efforts in China have tied poaching to illegal ivory products. A media campaign [63] using the slogan "no trade, no killing," for example, emphasizes that poaching is a direct consequence of demand for ivory products in China. Involving the public in efforts to eliminate illegal trading could dramatically reduce demand and poaching. Thus, programs combining in situ protection (i.e., anti-poaching) and devaluing of ivory could have large impacts on donations.

Improving the quality of elephant habitat is the second most supported strategy for elephant conservation, suggesting that many Chinese residents are concerned about the welfare of the elephants in the wild. Human encroachment and agricultural projects in Africa have eliminated large portions of elephants' habitats, forcing overcrowding—depriving them of living space and reproduction opportunities [64,65]. Droughts have also degraded vegetation and water supplies in Africa [66–68]. The cost of expanding protected areas and improving food and water supplies available to elephants is of enormous concern. At the same time, the increasing human populations in those nations put even greater pressure on existing ecosystems. African elephants tend to visit human communities located near their habitats due to the short supply of the vegetation. These competing pressures on humans and elephants inevitably lead to conflict [69]. Consequently, when protecting the survival of elephants in the wild, it is important to provide habitats for them that adequately meet their needs for food, water, and space to reproduce, which will simultaneously reduce conflicts (i.e., with local villagers and farmers), thereby developing sound coexistence [7,70]. Large-scale habitat restoration projects should garner a significant amount of support from the public in China, especially when the survival of a species is at stake. Adequate communication via the media and biodiversity education is recommended.

That higher income increases the likelihood that people will donate to support African elephant conservation is particularly enlightening. There has been longstanding concern that economic development in countries such as China would lead to greater demand for ivory products and more poaching [9]. Our results suggest that higher education and income levels may benefit African elephant conservation. Education helps increase residents' awareness of the plight of elephants in the wild, and greater income overall provides disposable income that can be directed to such causes of concern. Many residents in China (and elsewhere) are not aware of dwindling biodiversity around the world, have no intention of contributing to wildlife conservation, and still desire to buy wildlife products such as ivory without considering the legitimacy of their sources [29]. By providing information

about these issues to the public, it should bolster the willingness to contribute to the African elephant conservation efforts.

In detail, interesting geographic differences in WTD for African elephant conservation were noted. Respondents in Beijing and Fuzhou expressed a greater likelihood of donating and a higher WTD than respondents in Guangzhou. This may be due to the fact that, compared to respondents in Guangzhou, participants in Beijing had relatively higher education levels on average and participants in Fuzhou had relatively higher income levels. Shanghai has relatively fewer educational events and activities on elephant conservation, leading to lower interest in conservation in general. Guangzhou (in 2014) and Beijing (in 2015) had held large public events in which illegal ivory products were destroyed, but ivory seizures continued in Guangzhou, which might have offset the effectiveness of conservation events there. Our results indicate that outreach activities in China that address the importance of seizing illegal ivory products are likely to lead to greater interest in conservation measures and donations to projects. However, it should be stated that since respondents were asked solely to respond to WTD to African elephant conservation—via a single focus survey—if they are faced with serval other competing charity requests, including other wildlife conservation species, their WTD to the African elephant might decrease.

## 5. Conclusions

This study employed the choice experiment method to estimate Chinese residents' WTD to African elephant conservation. The results reveal that about two-thirds of the sample population, which was skewed somewhat toward higher levels of education and income than the general population, were willing to donate to efforts to protect African elephants. Intuitively, we further found that participants' WTD decreased with the size of the donation required. However, since even the highest donation level in the study of RMB 500 represented only about 0.3% of an average family's income, we believe there is significant, currently untapped potential for donations to elephant conservation by individuals in China. African elephant conservation strategies—anti-poaching, habitat enhancement, combating illegal ivory trading, and alleviating human–elephant conflicts—were all found to statistically significantly affect Chinese residents' WTD. Anti-poaching strategy attracted the highest WTD compared to the other three strategies.

Geographically, participants in Beijing and Fuzhou were more likely to donate to elephant conservation efforts than participants in Guangzhou. An NGO in China for conservation of elephants in Africa may lead to significant public donations. The foundation would act as a capital pool that would accept public donations and use them to fund projects for African elephant conservation. Supporting projects that combine multiple prongs of attack, such as anti-poaching measures and habitat enhancement in Africa and reductions in trading of illegal ivory in China may attract more donations. Outreach and educational activities may improve Chinese citizens' awareness of the problems associated with ivory and to feel compassionate towards the conservation efforts. Education about the dire circumstances for elephants in Africa and the bloody consequences of the ivory trade may be helpful to increase people's engagement in elephant conservation and may increase their WTD. As economic conditions in China improve, we expect that an even greater share of the country's people will be willing to donate to protect elephants and that the sizes of their donations could increase. Public donations from less developed countries may make a significant contribution to African elephant conservation in the future.

**Author Contributions:** Conceptualization, S.W. and Y.X.; methodology, Z.C. and Y.X.; software, S.W.; validation, Y.X.; formal analysis, S.W. and Y.X.; investigation, S.W. and Y.H.; writing—original draft preparation, S.W.; writing—review and editing, Y.X., Z.C. and G.T.C.; visualization, Y.H.; supervision, Y.X.; funding acquisition, Y.X. All authors have read and agreed to the published version of the manuscript.

**Funding:** The National Natural Science Foundation of China (grant number 71841147001).

**Acknowledgments:** The authors acknowledge the Center of Conservation Biology of the University of Washington for hosting the corresponding author as a Fulbright visiting scholar from December 2018 to October 2019. The authors are also grateful to two anonymous reviewers for their beneficial comments.

**Conflicts of Interest:** The authors declare no conflict of interest. The funder had no role in the design of the study, the collection, analyses, and interpretation of data, the writing of the manuscript, and the decision to publish the results.

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
