# Peer review of "Chinese Resident Preferences for African Elephant Conservation: Choice Experiment"

_diversity, doi:10.3390/d12120453_

Round 1

Reviewer 1 Report

General comments:

This work represents a much-needed study and I would like to congratulate the authors on their efforts. The manuscript is well formulated and the science sound. The topic is novel and very relevant to the increased protection of the African elephant. The increased funding demands to ensure the protection of elephants in Africa is evermore relevant since the Covid-19 pandemic. More studies such as these are needed. It would be interesting to conduct such studies in more countries and then do a transnational comparative analysis of the results.

Minor comments:

METHOD: Page 4, line 134-136: remove the ''of'' before the ''were''. 

DATA COLLECTION: Page 5, line 167-168: ''Choice experiments were included in these two parts''. The reader would benefit from more clarity on precisely to which of the five parts the choice experiments were being applied. I would first list the five parts and then add the sentence on which two parts of the five, the choice experiments were being applied to. So the sentence seems out of place.

DISCUSSION: Page 10, line 337-353: This whole paragraph would benefit from using more references to substantiate many of the statements that are made. There is a wealth of relevant literature available on various topics simply listed as statements.

Reviewer 2 Report

This is a very important topic, focusing on the funding of wildlife conservation programmes and the sustainability of conservation moving forward. The paper is quitewell written, with the exception of the methodology (in particular section 2.1) and some grammatical errors (see sample list below).

The introduction is quite good, although a broader examination of wildlife conservation in general and contributions from the public in developed countries could be included. L94-100 the paper follows the scientific convention of structure so this is unnecessary. A clearly stated aim/hypothesis would be better.

Section 2.1 is particularly hard to follow, and it is not clear to the reader at the end how the questionnaire was formatted, what the 32 management practices were, whether the respondents were to pick one prefered practice, pick more than one, rank them... I presume the models are correctly carried out but there is a need for clarity in the earlier sections to allow everything to fall into place.

If the choices for donation range from 0 to RMB500, how was a mean of RMB 641.25 achieved? This may be due to the poor explanation of the methods, but it is not clear at all.

The results are very limited by the selection of respondents, and the authors would have been better trying to make their survey more representative of the general population. There is still merit in the data, but it does lead to speculation. Based on the experience in developed countries, the estimate that 1.5% of the population will donate (10% of the 15% with bachelor degrees), seems to be optimistic. Respondents to the survey may be open to donating when faced with a single focus survey, but when faced with several other competing charity requests, including many other conservation causes, this will decrease even further. This is not considered.

There are several papers listed in the references that are not cited in the text and should be removed... Champ et al 2017, Cyranoski 2016, Johns 2010, Johnston et al 2017, Haab & McConnell 2002, Hanley et al 2002, MacDonald et al 2005, Novacek 2008, Saunders et al 2006, Sullivan 2000, Valasuk et al 2017.

Abstract L17-18 2/3 of a subsection (higher educated + income) is an unknown quantity without info on the size of this subsection in society

L45 annually

L52 to support

L53-54 is limited... is no enough to conserve

L71 increasing significantly

L72-73 knowledge on... is limited

L83 is unknown

L88-89 to four current African...

L107 This is the first use of WTP. Does this mean willingness to pay? Is this distinct from willingness to donate. If no, just use one, if yes, then clearly define each and explain how they differ.

L122-124 poor grammar, meaning lost

L135-136 Is there a word missing? Efficiency of what?

L236 - 238 This is discussion. References should rarely be used in results

L259 did not

L270 Is RMB 109.11/year the difference between your values and those of Lee and Preez? This should be rewritten for clarity. Also it is discussion.

Round 2

Reviewer 2 Report

Although the authors have made an effort to improve the writing and to respond to previous comments, the changes made are relatively minor, when a major revision was required, in particular of the methodology. The main concern about section 2.1 has not been addressed, it is still not clear how the questionnaire was formatted, or what the 32 management practices were - and how they correspond to the broader conservation approaches (anti-poaching, trade in ivory etc). The language in one paragraph has been improved but that is not sufficient.

With regards them disagreeing with my query as to how a mean of RMB 641.25 can develop from choices ranging from 0 to RMB 500, I don't doubt the result. I said it was not clear how it was derived due to the poor explanation of the methods. So that has not been addressed.

The authors note that this is a pilot study, and publication might be better to wait for the full study to be carried out.
